# Ga_2_Te_3_-Based Composite Anodes for High-Performance Sodium-Ion Batteries

**DOI:** 10.3390/ma15186231

**Published:** 2022-09-08

**Authors:** Vo Pham Hoang Huy, Il Tae Kim, Jaehyun Hur

**Affiliations:** Department of Chemical and Biological Engineering, Gachon University, Seongnam 13120, Gyeonggi, Korea

**Keywords:** Ga_2_Te_3_, Ga_2_Te_3_–TiO_2_–C, anodes, Na ion, sodiation/desodiation

## Abstract

Recently, metal chalcogenides have received considerable attention as prospective anode materials for sodium-ion batteries (SIBs) because of their high theoretical capacities based on their alloying or conversion reactions. Herein, we demonstrate a gallium(III) telluride (Ga_2_Te_3_)-based ternary composite (Ga_2_Te_3_–TiO_2_–C) synthesized via a simple high-energy ball mill as a great candidate SIB anode material for the first time. The electrochemical performance, as well as the phase transition mechanism of Ga_2_Te_3_ during sodiation/desodiation, is investigated. Furthermore, the effect of C content on the performance of Ga_2_Te_3_–TiO_2_–C is studied using various electrochemical analyses. As a result, Ga_2_Te_3_–TiO_2_–C with an optimum carbon content of 10% (Ga_2_Te_3_–TiO_2_–C(10%)) exhibited a specific capacity of 437 mAh·g^−1^ after 300 cycles at 100 mA·g^−1^ and a high-rate capability (capacity retention of 96% at 10 A·g^−1^ relative to 0.1 A·g^−1^). The good electrochemical properties of Ga_2_Te_3_–TiO_2_–C(10%) benefited from the presence of the TiO_2_–C hybrid buffering matrix, which improved the mechanical integrity and electrical conductivity of the electrode. This research opens a new direction for the improvement of high-performance advanced SIB anodes with a simple synthesis process.

## 1. Introduction

In the last few decades, lithium-ion batteries (LIBs) have been utilized as an effective alternative to unsustainable fossil fuels in energy storage systems such as portable electronic devices and electric vehicles [1,2,3,4,5,6,7,8,9,10,11,12,13,14]. Nevertheless, by virtue of the limited reserves and the high cost of Li, much attention has been drawn to developing alternative secondary batteries to overcome these issues [15]. Sodium-ion batteries (SIBs) are reputed as one of the most viable secondary batteries among many next-generation batteries because of their many similarities to LIBs and the abundance of Na on earth [16,17,18,19]. Because of the chemical similarity between Na and Li, the Na storage mechanism of anode materials is similar to that of LIB systems, including intercalation/deintercalation, conversion, and alloying/dealloying reactions. Therefore, considerable attempts have been dedicated to finding suitable anode materials for SIBs. However, the slow reaction kinetics owing to the large ionic radius (1.03 nm for Na^+^ relative to 0.75 nm for Li^+^) causes low cycling stability and rate capability or even complete electrochemical inactivity. Thus, the improvement of desirable anode materials for high-performance SIBs is urgently required for expanding their practical application scope. Recently, alloy-based materials have received considerable attention as SIB anode materials owing to their high theoretical capacities (Na–Si: 955 mAh·g^−1^, Na–Ge: 368 mAh·g^−1^, Na–Sn: 845 mAh·g^−1^, Na–Pb: 486 mAh·g^−1^) [20]. However, similar to LIBs, the large volume change of these materials significantly restricts the long-term cycling of SIBs.

At present, there are numerous potential anodes studied for SIBs, which can be classified in accordance with the reaction mechanism: intercalation (carbon-based materials) [21,22,23], conversion (sulfides and oxides) [24,25,26], and alloying reaction (Sb, Sn, and P) [27,28,29]. Among these, chalcogenide materials (particularly S- and Se-based materials) have received significant attention as they undergo sequential alloying and conversion reactions, resulting in high capacities [30,31,32]. Te, another chalcogen group element, has recently been recognized as an effective substitute for S and Se by virtue of the high theoretical Na storage capacity (420 mAh·g^−1^) based on alloying with two Na ions (Na_2_Te) [33]. Furthermore, the high density of Te (6.23 g·cm^–3^) results in a high theoretical volumetric capacity of 2621 mAh·cm^–3^, which is comparable to that of S (3468 mAh·cm^–3^) and Se (3252 mAh·cm^–3^). Furthermore, Te has superior electrical conductivity (ca. 1.9 × 10^−2^ S·cm^– 1^) when compared to other nonmetallic chalcogenides (S (ca. 5.1 × 10^–16^ S·cm^–1^) and Se (ca. 1.1 × 10^–4^ S·cm^–1^) [34]. In line with this, although a Ga_2_S_3_-based composite anode was previously used as the anode for SIB, the rate performance was not satisfactory due to the low electronic conductivity of S [35]. In contrast, Te-based composite electrodes have achieved good rate performance and high capacity retention thanks to the high electronic conductivity of Te, as shown in Appendix A [36,37,38,39,40,41,42,43,44]. Despite these great characteristics, the study of the Te-based SIB electrode mechanism has so far been rare [39,45,46,47,48].

Ga-based materials, such as Ga oxide/sulfide anodes, have a large theoretical capacity (682–1591 mAh·g^−1^), innate self-healing capability, and a high tolerance against volume change [49,50,51,52]. Ga-based materials have recently emerged as potential electrode materials because of their unique self-healing properties based on the low melting temperature of Ga (29.9 °C). The intermediate liquid Ga formed during sodiation increases the tolerance of volume expansion of active materials, which significantly contributes to the cycling stability [53]. For instance, a composite including reduced graphene oxide and gallium oxide nanosheets (Ga_2_O_3_ NSs/rGO) by Yang et al. provided a steady capacity of 555 mAh·g^−1^ at 0.1 A·g^−1^ [54], whereas a template-derived Ga_2_S_3_ nanorod anode could obtain a discharge capacity of 476 mAh·g^−1^ at 0.4 A·g^−1^ [55]. Using in situ microscopy, Wu et al. investigated the self-healing properties of a liquid metal Ga-alloy during the charge/discharge process [56]. Considering the aforementioned advantages of Te and Ga, gallium telluride alloys (i.e., Ga_x_Te_y_) are expected to be great candidate anode materials for SIBs.

In this work, a Ga_2_Te_3_-based composite electrode (Ga_2_Te_3_–TiO_2_–C) was successfully synthesized utilizing a simple solid-state high-energy ball milling (HEBM) method and studied as a potential SIB anode material. The feasibility of the Ga_2_Te_3_–TiO_2_–C anode for SIBs was investigated through galvanostatic measurements, differential capacity analysis, and electrochemical impedance spectroscopy (EIS). Furthermore, the reaction mechanism of Ga_2_Te_3_–TiO_2_–C anode during sodiation/desodiation was first investigated via ex situ X-ray diffraction (XRD) analysis. In addition, the best C content (10 wt.%) in the Ga_2_Te_3_–TiO_2_–C composite was derived through various electrochemical tests. The high cycling and rate performances of Ga_2_Te_3_–TiO_2_–C(10%) obtained in this study are superior or equivalent to those of the most recent chalcogenide-based electrodes in SIBs.

## 2. Experiment

### 2.1. Material Synthesis

The crystalline Ga_2_Te_3_ alloy was synthesized using a simple solid-state HEBM. First, commercial powder of Ti (325 mesh, 99.99%, Alfa Aesar, Haverhill, MA, USA), Ga_2_O_3_ (99.99%, Sigma Aldrich, St. Louis, MO, USA), Te (99.8%, Alfa Aesar) (where the ratio is 6:2:3), and ZrO_2_ balls were placed in the milling bowl (powder mixture: ball = 1:20, *w*/*w*). Under an Ar atmosphere, the powder compound was milled for 10 h at 300 rpm. Subsequently, the obtained powder (Ga_2_Te_3_–TiO_2_) was manually mixed with acetylene carbon black powder (C) (99.9+%, bulk density: 170–230 g·L^−1^, S.A.: 75 m^2^·g^−1^, Alfa Aesar) in mass ratios of 9:1 (Ga_2_Te_3_–TiO_2_–C(10%)), 8:2 (Ga_2_Te_3_–TiO_2_–C(20%)), and 7:3 (Ga_2_Te_3_–TiO_2_–C(30%)). The mixtures were ball-milled under the same condition as the first milling. The mechanochemical reaction route for preparing the Ga_2_Te_3_–TiO_2_–C composite is outlined below.
First step: 2Ga_2_O_3_ + 6Te + 3Ti → 2Ga_2_Te_3_ + 3TiO_2_ (Ga_2_Te_3_–TiO_2_),(1)
Second step: Ga_2_Te_3_–TiO_2_ + C → Ga_2_Te_3_–TiO_2_–C.(2)

### 2.2. Material Characterization

The morphology and crystallinity of Ga_2_Te_3_–TiO_2_ and Ga_2_Te_3_–TiO_2_–C were characterized by employing XRD (D/MAX–2200 Rigaku, Tokyo, Japan) with Cu Kα (λ = 1.54 A) radiation at a scan rate of 2°·min^−1^, as well as energy-dispersive X-ray spectroscopy (EDXS), scanning electron microscopy (SEM, Hitachi S4700, Tokyo, Japan), and high-resolution transmission electron microscopy (HRTEM, JEOL JEM-2100F, Tokyo, Japan). The mechanism of phase change state in the electrode during Na ion reaction was performed through ex situ XRD.

### 2.3. Electrochemical Measurements

To assess the electrochemical performance of the electrode, the coin-typed cells were assembled in an Ar-filled glovebox with Na foil acting as the counter electrode and polyethylene as the separation membrane. The SIB electrolyte was 1 M NaClO_4_ in propylene carbonate/ethylene carbonate (1:1 by *v*/*v*) with 5% fluoroethylene carbonate. An electrode was prepared using a 7.0:1.5:1.5 (*w*/*w*) combination of the active material, conductive carbon (Super-P, 99.9%, Alfa Aesar), and poly(acrylic acid) (PAA, Mw 450,000, Sigma Aldrich) binder. Then, electrodes were coated on Cu foil using a doctor blade and dried overnight in a vacuum oven at 70 °C (a typical mass loading of 1.0−1.5 mg on a Cu foil diameter of 12.5 mm). Using a battery-testing device (WBCS3000, WonATech, Seoul, Korea), the electrochemical performance of Ga_2_Te_3_–TiO_2_–C was assessed. For a voltage range of 0.01–2.5 V (vs. Na/Na^+^), the galvanostatic charge–discharge (GCD) profile was investigated. Using cyclic voltammetry (CV) measurements with a scanning rate of 0.1 mV·s^−1^, the electrochemical reactions of the electrodes with Na^+^ were evaluated. A battery cycler (WBCS3000, WonATech) was used to measure the rate capability at current densities of 0.1, 0.5, 1, 3, 5, and 10 A·g^−1^. The EIS was measured using a ZIVE MP1 (WonaTech) analyzer in the frequency range of 100 kHz–100 mHz at an AC amplitude of 10 mV.

## 3. Results and Discussion

The XRD results of the as-prepared Ga_2_Te_3_–TiO_2_ following the HEBM process are shown in Figure 1a. The peaks at 26.2°, 30.3°, 43.4°, 51.4°, 53.8°, 63.0°, 69.4°, and 79.5° corresponded to the (111), (200), (220), (311), (222), (400), (331), and (422) planes of Ga_2_Te_3_, respectively, indicating the formation of monoclinic Ga_2_Te_3_. The relatively small peaks observed at 33.3°, 48.5°, 67.9°, and 79.0° matched the (311), (020), (621), and (424) planes of TiO_2_, respectively. The presence of amorphous C decreased the crystallinity of Ga_2_Te_3_ and TiO_2_ in Ga_2_Te_3_–TiO_2_–C (Appendix A) [57]. The absence of impurity peaks for the precursor components (Ga_2_O_3_, Ti, and Te) in Ga_2_Te_3_–TiO_2_ indicated the complete conversion of the raw materials to the target product via a solid-state reaction. Nevertheless, the existence of small diffraction peaks near 2θ = 22.9°, 35.1°, and 37.5° were associated with the formation of GaTe (PDF#33-0571) and Ga_2_Te_5_ (PDF#45-0954), which are different forms of gallium telluride alloy (as shown in Appendix A). Given the advantages of gallium telluride alloy (i.e., Ga_x_Te_y_), they are also expected to be good candidate anode materials for SIBs; thus, the existence of different forms of minor Ga_x_Te_y_ (namely, Ga_2_Te_5_ and GaTe) did not negatively affect the electrochemical performance of the Ga_2_Te_3_–TiO_2_–C composite (capacity still in the range of 682−1591 mAh·g^−1^). The EDXS showed that the four component elements of the composite in the electrode and the relative content of elements were reasonable with the stoichiometric ratio of Ga_2_Te_3_ and TiO_2_ (Figure 1b), indicating that the Ga_2_Te_3_-based composite was synthesized successfully.

Morphological and structural analyses of Ga_2_Te_3_–TiO_2_–C(10%) were conducted, including SEM, HRTEM, and EDXS, as indicated in Figure 2. According to SEM images (Figure 2a,b), the Ga_2_Te_3_–TiO_2_–C(10%) particle size ranged from sub-micrometers to a few micrometers. The HRTEM images (Figure 2c and Appendix A) revealed crystalline lattice spacings of 0.340, 0.294, and 0.208 nm, which corresponded to the (111), (200), and (220) crystal planes of Ga_2_Te_3_, respectively, and 0.311 nm, which corresponded to the (002) plane of TiO_2_. Additionally, amorphous C, which was anticipated to serve as a buffering network for the active material, formed surrounding Ga_2_Te_3_ and TiO_2_. The scanning transmission electron microscopy image with EDXS mapping analysis (Figure 2d and Appendix A) showed a uniform distribution of each element (Ga, Te, Ti, O, and C) in the Ga_2_Te_3_–TiO_2_–C(10%). Furthermore, the SEM–EDXS analysis results (Appendix A) of Ga_2_Te_3_–TiO_2_ with different content of C showed C concentrations almost identical to their theoretical values. Additionally, the stoichiometric ratio of the constituent elements was nearly identical to the theoretical values, according to a quantitative analysis of the EDXS result. In the EDS spectrum of G_2_Te_3_–TiO_2_, Te was determined to be 25%. However, in the presence of 10% carbon content, the distribution of Te gradually decreased (specifically to 18%, as shown in EDS spectrum of Ga_2_Te_3_–TiO_2_–C(10%) in Appendix A); thus, the distribution of Te content in the TEM elemental mapping was relatively less.

The Na ion storage characteristics of the Ga_2_Te_3_–TiO_2_–C electrode were studied using a half-cell form with Na metal as the counter electrode (Figure 3). The GCD voltage profiles of Ga_2_Te_3_–TiO_2_–C(10%), Ga_2_Te_3_–TiO_2_–C(20%), and Ga_2_Te_3_–TiO_2_–C(30%) for SIBs are shown in Figure 3a and Appendix A. The first discharge/charge capacities of Ga_2_Te_3_–TiO_2_–C(10%), Ga_2_Te_3_–TiO_2_–C(20%), and Ga_2_Te_3_–TiO_2_–C(30%) were 599/414, 550/357, and 462/279 mAh·g^−1^, respectively, which corresponded to initial coulombic efficiencies (ICEs) of 69.1%, 64.9%, and 60.4%, respectively. The poor reversibility between the first and second cycles was because of the SEI layer formation in the first cycle. This poor reversion was well documented in previous studies [58,59,60]. The large capacity difference between the first and second cycle indicated the large irreversible capacity contribution from the SEI layer. However, the reversibility of the electrode (Ga_2_Te_3_–TiO_2_–C (10%) was rapidly enhanced after the second cycle, which could be confirmed by the change in coulombic efficiency (Appendix A). According to the EDXS results (Appendix A) and computed theoretical capacities of the separate components (Appendix A), the capacity contributions of C and TiO_2_ to Ga_2_Te_3_–TiO_2_–C(10%) were estimated to be 13% and 22%, respectively. Furthermore, the roles of C and active material were examined (as shown in Appendix A). Ga_2_Te_3_–TiO_2_ achieved a high initial capacity (606 mAh·g^−1^), but its capacity gradually decreased due to the instability of the electrode structure without buffering C. Moreover, the electrode with only a buffering matrix (TiO_2_–C) showed very low electrochemical efficiency, close to the theoretical capacity (116 mAh·g^−1^) (Appendix A). The low-capacity contribution of the TiO_2_–C (~35%) indicated its main role as a buffering matrix. Due to interfacial Na ion storage and electrolyte breakdown, the measured capacities of Ga_2_Te_3_–TiO_2_–C(10%) and Ga_2_Te_3_–TiO_2_ in the SIBs were higher than their theoretical capacities (336 and 333 mAh·g^−1^, respectively, as computed in Appendix A). The change in the reversible capacity of Ga_2_Te_3_–TiO_2_–C for the SIBs was studied using the CE (Appendix A) and DCP test of the first 300 cycles (Appendix A). The CE of Ga_2_Te_3_–TiO_2_–C(10%) reached ~99.82% after 150 cycles, slightly decreased, and then stabilized at 98.5% after 300 cycles. The DCP analysis revealed that, for 250 cycles, the main oxidation (at ~0.16, ~1.27, and ~1.42 V) and reduction (at ~0.79 and ~1.58V) peaks remained stable before becoming wider and shifting. However, this polarization had an almost negligible effect on sodiation/desodiation. The reversible capacity of Ga_2_Te_3_–TiO_2_–C(10%) was 436.6 mAh·g^−1^ (capacity retention (CR) of 97.7%) after 300 cycles at 100 mA·g^−1^, which was greater than those of Ga_2_Te_3_–TiO_2_–C(20%) (323.8 mAh·g^−1^) and Ga_2_Te_3_–TiO_2_–C(30%) (264.9 mAh·g^−1^) (Figure 3b). As shown in Appendix A, although some aggregated particles were observed, the Ga_2_Te_3_–TiO_2_–C(10%) electrode morphology was generally well maintained after 300 cycles. This is because of the presence of TiO_2_–C, which effieicntly stabilized the electrode structure and mitigated the significant volume variation. In addition, in EDS spectra after 300 cycles, the composition of the Ga_2_Te_3_ composite electrode was not significantly changed without impurities (Appendix A). This further proved the stability and good retention of the electrode after the electrochemical reaction. At 500 mA·g^−1^ (Figure 3c), the reversible capacity of Ga_2_Te_3_–TiO_2_–C(10%) slightly increased until 200 cycles, followed by a gradual decrease. The capacity variation depends on the variation of the redox peaks, in which the oxidation and reduction peaks gradually rise with the cycling, leading to a decrease in polarization and an increase in capacity. In contrast, the oxidation and reduction peaks gradually decrease with the increase in cycling, resulting in a reduced capacity due to the increase in polarization [10,61,62]. This trend was also shown in the DCP analysis (Appendix A) and CE variation (Appendix A). The magnitudes of the reduction (at 0.59 and 1.48 V) and oxidation (at 0.16, 1.27, and 1.69 V) peaks gradually raised over 200 cycles, with a reduction in polarization (Appendix A), and then reduced after 200 cycles, with a rise in polarization (Appendix A). At 100 and 500 mA·g^−1^, the fluctuation of the DCP profile was examined as a function of the cycle number (Appendix A). The DCP curves of the Ga_2_Te_3_–TiO_2_–C(10%) electrode showed that the overall intensity of the redox peaks was generally stable as the cycle number increased to 300 at 100 mA·g^−1^. At 500 mA·g^−1^, the overall magnitudes of the redox peaks increased up to 250 cycles and then decreased with an increase in polarization. Despite the decrease in capacity after 250 cycles, the overall capacity of Ga_2_Te_3_–TiO_2_–C(10%) was still the highest over 500 cycles, reaching 204 mAh·g^−1^ after 500 cycles with a CR of 76.4%. Appendix A shows a comparison of the CE variations in Ga_2_Te_3_–TiO_2_–C with varying C contents at 100 and 500 mA·g^−1^. Appendix A (at 100 mA·g^−1^) and Appendix A (at 500 mA·g^−1^) provide summaries of the detailed CE values for the electrodes throughout the first 10 cycles. As shown in Appendix A, the ICE of the Ga_2_Te_3_–TiO_2_–C(10%) electrode was slightly higher (69.2%) than that of the Ga_2_Te_3_–TiO_2_–C(20%) (ICE = 64.8%) and Ga_2_Te_3_–TiO_2_–C(30%) electrodes (ICE = 60.5%). Then, after 10 cycles, the CE of the Ga_2_Te_3_–TiO_2_–C(10%) electrode marginally increased and reached the highest among the three various electrodes. This tendency was also discovered at 500 mA·g^−1^ (Appendix A). After the first cycle, the high CE of the Ga_2_Te_3_–TiO_2_–C(10%) electrode showed a high degree of sodiation/desodiation reversibility. The CV curves of the Ga_2_Te_3_–TiO_2_–C(10%) electrode for the first five cycles in the voltage range of 0.005–2.5 V vs. Na/Na^+^ are shown in Figure 3d. A large reduction peak was observed at 1.37 V during the first discharge process, which denoted the intercalation of Na into Ga_2_Te_3_ to form Na_2_Te and Ga. The reaction between Ga and Na to generate NaGa_4_ was attributed to being responsible for the peak at 0.52 V. Thus, Na_2_Te and NaGa_4_ were the final products after the discharge step was complete. In the charge step, two oxidation peaks were noticed at 0.92 and 1.72 V. The first peak was the result of Na being completely excluded, turning NaGa_4_ into Ga. Then, Ga intruded into Na_2_Te to form Ga_2_Te_3_ when the anode was charged to 1.72 V. The ex situ investigations concern a thorough analysis of this phase transition. After the second cycle, the curves nearly overlapped, indicating the excellent stability and reversibility of Ga_2_Te_3_–TiO_2_–C(10%). The CV curves of Ga_2_Te_3_–TiO_2_–C(20%) and Ga_2_Te_3_–TiO_2_–C(30%) were almost identical to that of Ga_2_Te_3_–TiO_2_–C(10%), with a similar level of cyclic stability after the second cycle (Appendix A). In addition, the electrochemical performance of Ga_2_Te_3_–TiO_2_ was examined (Appendix A). The GCD profiles of Ga_2_Te_3_–TiO_2_ presented initial charge/discharge capacities of 606/426 mAh·g^−1^, corresponding to an ICE of 70.2%, which is higher than that of the Ga_2_Te_3_-based composite with different C contents. Despite this high ICE of Ga_2_Te_3_–TiO_2_, the capacity gradually decreased with the increase in cycle number, and reached 309 mAh·g^−1^ after 30 cycles, with a capacity retention of 67%. This is much lower than the Ga_2_Te_3_–TiO_2_ electrode with various carbon contents. In addition, the CV curves did not overlap in the first five cycles. Therefore, the presence of C clearly stabilized the electrode structure, leading to the enhanced electrochemical performance. The rate performances (Figure 3e) and normalized capacity retentions (Figure 3f) of the electrodes were determined. At 0.1, 0.5, 1.0, 3.0, 5.0, and 10.0 A·g^−^^1^, the average specific capacities of Ga_2_Te_3_–TiO_2_–C(10%) were 455, 408, 374, 348, 321, and 318 mAh·g^−^^1^, respectively (Figure 3e), which were considerably greater than those of Ga_2_Te_3_–TiO_2_–C(20%) and Ga_2_Te_3_–TiO_2_–C(30%). Surprisingly, even at 10 A·g^−1^, Ga_2_Te_3_–TiO_2_–C(10%) had a CR of up to 96% (Figure 3f). Additionally, Ga_2_Te_3_–TiO_2_–C(10%) demonstrated a high rate performance when the discharge rate was reduced from 10 A·g^−1^ to 0.1 A·g^−1^, resulting in high CR (99.3%).

The reaction mechanism during the first sodiation/desodiation process of the Ga_2_Te_3_–TiO_2_–C(10%) electrode was investigated using ex situ XRD (Figure 4a,b). Peaks corresponding to Na_2_Te and Ga were observed at a discharge voltage of 1.37 V (D: 1.37 V). When the electrode was fully discharged (D: 5 mV), NaGa_4_ peaks were observed and Na_2_Te peaks remained. The NaGa_4_ phase partly disappeared when the electrode was charged to 0.92 V (C: 0.92V). In a charging state of 1.72 V, the Na_2_Te phase partly disappeared, Ga was observed, and NaGa_4_ completely disappeared. Only the peaks corresponding to Ga_2_Te_3_ were observed again when the electrode was fully charged to 2.5 V (C: 2.5 V). Ga_2_Te_3_ undergoes the following structural changes during sodiation/desodiation:
❖1st discharge
Intercalation stageGa_2_Te_3_ + xNa^+^ + xe^−^ → Na_x_Ga_2_Te_3_ (2.5−1.37 V). (i)Conversion stageNa_x_Ga_2_Te_3_ + (6^−^) Na^+^ + (6^−^x)e^−^ → 3Na_2_Te + 2Ga (1.37−0.52 V). (ii)Alloy stage4Ga + Na^+^ + e^−^ → NaGa_4_ (0.52−0.005 V). (iii)❖1st charge
De-alloy stageNaGa_4_ → 4Ga + Na^+^ + e^-^ (0.005−0.92 V). (iv)De-conversion stage3Na_2_Te + 2Ga → Li_x_Ga_2_Te_3_ + (6^−^x) Na^+^ + (6^−^x)e^−^ (0.92−1.72 V). (v)De-intercalation stageNa_x_Ga_2_Te_3_ → Ga_2_Te_3_ + xNa^+^ +xe^−^ (1.72−2.5 V). (vi)


It is noteworthy that, after the first cycle, the Ga_2_Te_3_ phase (major peaks at 53.8°, 69.4°, and 71.5°) was completely recovered without any impurity peaks, showing a highly reversible interaction of Ga_2_Te_3_ with Na ions. The alloying/dealloying and conversion mechanism of the Ga_2_Te_3_ electrode during charge/discharge is shown by the ex situ XRD results, as schematically depicted in Figure 4c.

For the first, fifth, and 20th cycles, the EIS profiles of the Ga_2_Te_3_–TiO_2_–C(10%), Ga_2_Te_3_–TiO_2_–C(20%), and Ga_2_Te_3_–TiO_2_–C(30%) electrodes were obtained (Figure 5). The simplified equivalent circuit shown in Figure 5d includes the electrolyte resistance (R_b_), charge-transfer resistance (R_ct_), SEI layer resistance (R_SEI_), interfacial double-layer capacitance (C_dl_), Warburg impedance (Z_w_), and constant phase element (C_PE_). The R_ct_ at the electrode–electrolyte interface is denoted by compressed semicircles in the mid-frequency region of the Nyquist plot. For all the electrodes, R_ct_ gradually decreased as the cycle number increased from 1 to 20. Ga_2_Te_3_–TiO_2_–C(10%) exhibited the lowest value of R_ct_ after 20 cycles (Appendix A), demonstrating the most facile Na ion transportation, which led to the highest Na storage performance.

Currently, there are only a few reports on Ga-based or Te-based anodes for SIBs. However, chalcogenide materials (In_2_S_3_, Sb_2_Se_3_, etc.) have high specific capacities when they undergo sequential conversion and alloying reactions owing to their unique properties. A comparison of the performances of Ga_2_Te_3_–TiO_2_–C(10%) and other chalcogenide materials demonstrated the high potential of the Ga_2_Te_3_-based composite electrodes for future applications (Table 1).

## 4. Conclusions

We demonstrated a Ga_2_Te_3_-based composite as a prospective anode material for SIBs. The Ga_2_Te_3_–TiO_2_·C(10%) anode achieved a high reversible capacity of 437 mAh·g^−1^ after 300 cycles at 0.1 A·g^−1^, as well as a high rate capability (CR of 96% at 10 A·g^−1^ relative to 0.1 A·g^−1^). The nanoconfined Ga_2_Te_3_ crystallites embedded within an electrically conductive TiO_2_–C hybrid matrix effectively accommodated the Ga_2_Te_3_ particle volume variation and avoided the agglomeration of Ga during electrochemical reactions. In addition, Na ion diffusion kinetics and mechanical stability were enhanced by this beneficial morphology, thereby achieving high capacity and long-term cycling performance These findings offer a new direction toward the development of high-performance SIBs with long cycle lifetimes and expansion of the Ga- and Te-based materials in other electrochemical energy storage systems.

## Figures and Tables

**Figure 1 materials-15-06231-f001:**
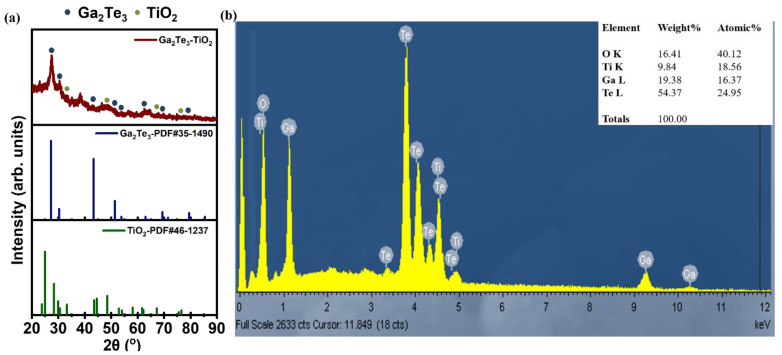
(**a**) XRD pattern of Ga_2_Te_3_–TiO_2_; (**b**) EDX spectrum of Ga_2_Te_3_–TiO_2_.

**Figure 2 materials-15-06231-f002:**
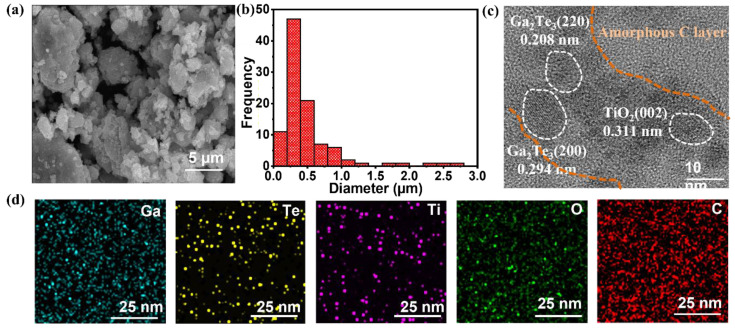
(**a**) SEM image; (**b**) size distribution; (**c**) HRTEM image; (**d**) EDXS elemental mappings of Ga, Te, Ti, O, and C in Ga_2_Te_3_–TiO_2_–C(10%).

**Figure 3 materials-15-06231-f003:**
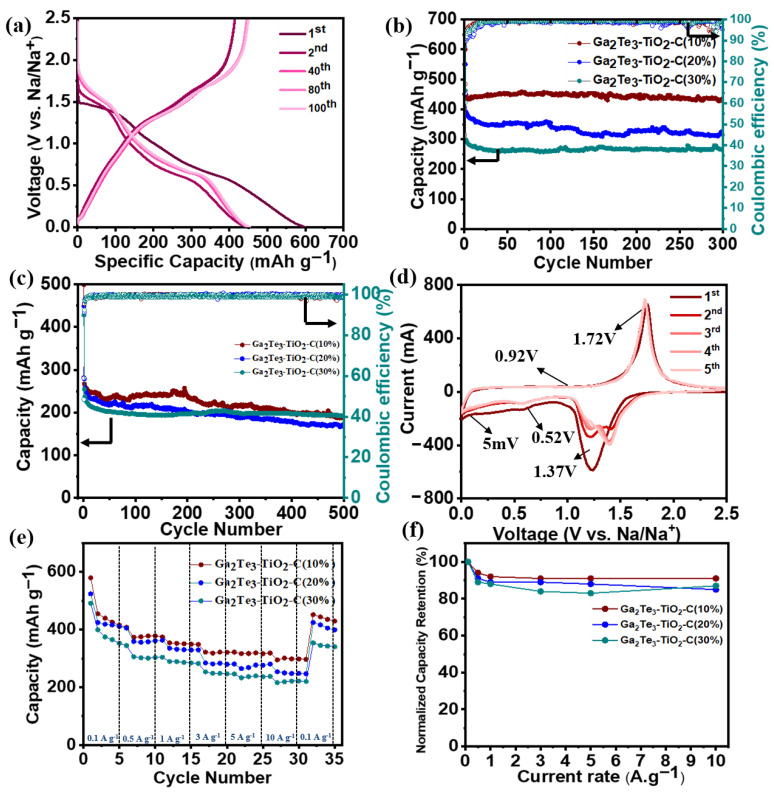
Electrochemical performances of Ga_2_Te_3_–TiO_2_–C composites for SIBs: (**a**) GCD profiles of Ga_2_Te_3_–TiO_2_–C(10%) at 100 mA·g^−1^; (**b**) cycling performances of composite at 100 mA·g^−1^ and (**c**) 500 mA·g^−1^; (**d**) CV curves of Ga_2_Te_3_–TiO_2_–C(10%); (**e**) rate capabilities of Ga_2_Te_3_–TiO_2_–C composites; (**f**) capacity retention of Ga_2_Te_3_–TiO_2_–C composites from 0.1 to 10 A·g^−1^.

**Figure 4 materials-15-06231-f004:**
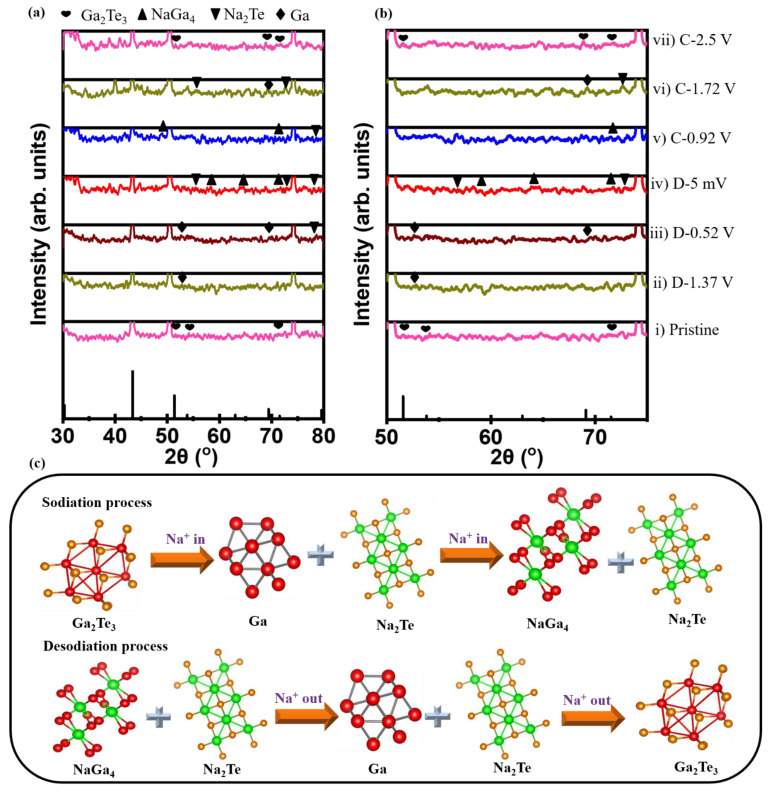
(**a**,**b**) Ex situ XRD patterns obtained at selected cutoff potentials in initial sodiation/desodiation process; (**c**) schematic of electrochemical reaction mechanism of Ga_2_Te_3_–TiO_2_–C(10%) electrode during cycling.

**Figure 5 materials-15-06231-f005:**
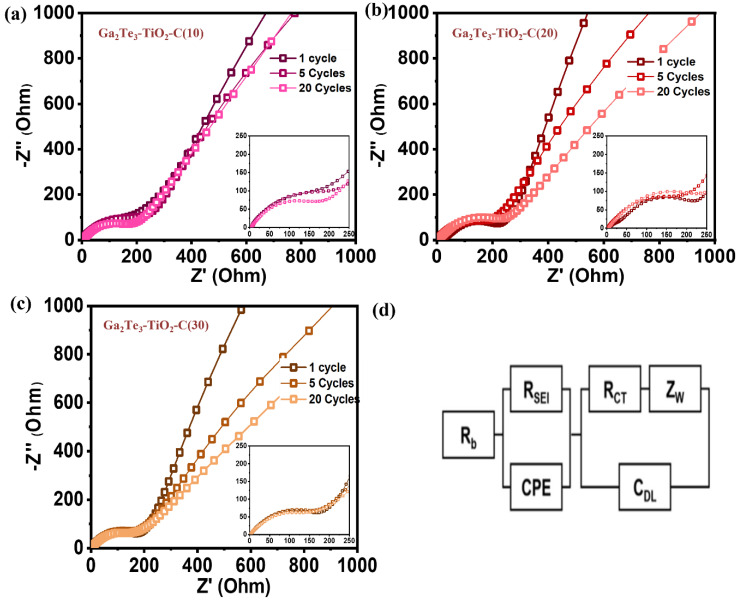
EIS-based Nyquist plots for (**a**) Ga_2_Te_3_–TiO_2_–C(10%), (**b**) Ga_2_Te_3_–TiO_2_–C(20%), and (**c**) Ga_2_Te_3_–TiO_2_–C(30%) after one, five, and 20 cycles; (**d**) equivalent circuit.

**Table 1 materials-15-06231-t001:** Comparison of performances of chalcogenide-based anodes for SIBs.

Anode	Cycling Performance	Rate Capability	Synthesis Method	Ref.
Ga_2_S_3_	476 mAh·g^−1^ after 100 cycles at 0.4 A·g^−1^	283 mAh·g^−1^ at 2 A·g^−1^	Vapor thermal annealing	[55]
Ga_2_S_3_–C	385 mAh·g^−1^ after 200 cycles at 0.1 A·g^−1^	94 mAh·g^−1^ at 2.0 A·g^−1^	Sulfuration process	[35]
Sb_2_Se_3_/C	485.2 mAh·g^−1^ after 100 cycles at 0.2 A·g^−1^	237.9 mAh·g^−1^ at 2.0 A·g^−1^	Hydrothermal process	[31]
Sb_2_S_3_–rGO	537 mAh·g^−1^ after 70 cycles at 0.1 A·g^−1^	290 mAh·g^−1^ at 3.2 A·g^−1^	Ultrasonication method	[63]
Sb_2_S_3_–C	455.8 mAh·g^−1^ after 100 cycles at 0.1 A·g^−1^	263 mAh·g^−1^ at 1.0 A·g^−1^	Modified natural stibnite ore	[64]
Sb_2_S_3_@SnS@C	437 mAh·g^−1^ after 100 cycles at 1 A·g^−1^	448 mAh·g^−1^ at 5.0 A·g^−1^	Hydrothermal method	[65]
Sb_2_S_3_	384 mAh·g^−1^ after 50 cycles at 0.2 A·g^−1^	239 mAh·g^−1^ at 5.0 A·g^−1^	Hydrothermal method	[66]
In_2_S_3_/C	372 mAh·g^−1^ after 200 cycles at 0.5 A·g^−1^	236 mAh·g^−1^ at 2.0 A·g^−1^	Electrospinning process	[67,68]
Co_3_Se_4_@C	449 mAh·g^−1^ after 20 cycles at 0.1 A·g^−1^	328 mAh·g^−1^ at 5.0 A·g^−1^	Annealing process	[69]
Fe_3_Se_4_@C	439 mAh·g^−1^ after 25 cycles at 0.05 A·g^−1^	-	Electrospinning process	[70]
Bi_2_Te_3_	364 mAh·g^−1^ after 1200 cycles at 5 A·g^−1^	339 mAh·g^−1^ at 10 A·g^−1^	Chemical reduction method	[36]
SbTe–C	421 mAh·g^−1^ after 200 cycles at 0.1 A·g^−1^	413 mAh·g^−1^ at 1 A·g^−1^	Ball milling	[71]
Ga_2_Te_3_–TiO_2_–C	437 mAh·g^−1^ after 300 cycles at 0.1 A·g^−1^	318 mAh·g^−1^ at 10 A·g^−1^	Ball milling	This work

## Data Availability

Not applicable.

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
