# Peer review of "Ga2Te3-Based Composite Anodes for High-Performance Sodium-Ion Batteries"

_materials, 2022, doi:10.3390/ma15186231_

Round 1

Reviewer 1 Report

In this work, the authors synthesized gallium (III) telluride (Ga2Te3)-based ternary composite (Ga2Te3-TiO2-C) via a simple high-energy ball mill, and employed as anode for SIBs. The Ga2Te3-TiO2-C sample with an optimum carbon content of 10% (Ga2Te3-TiO2-C (10%)) delivers a capacity of 437 mAh g-1 at 100 mA g-1 after 300 cycles. However, this manuscript still exists some mistakes and needs more deep analysis, so based on this consideration, I recommend major revision for publication in Materials.

1. In Figure 1a, the authors claimed that the diffraction peaks are all indexed to Ga2Te3 and TiO2 without impurities, however, an obvious diffraction peak observed at 37.5° can not match with the above target phases. So, the composite exists some impurities or the authors should index this diffraction peak.

2. In Figure 2d of EDS Mapping images, the authors should provide the origin SEM image to illustrate the well distribution of the elements.

3. The C content of Ga2Te3-TiO2-C samples should be confirmed by more characterizations.

4. The authors should provide the sodium storage properties of Ga2Te3-TiO2 sample without adding carbon to illustrate the influence of carbon component in Ga2Te3-TiO2-C composite

5. Figure 4a and 4b, the peak intensity in ex-situ XRD pattern of Ga2Te3-TiO2-C sample is weak, and it is hard to confirm the interphases. Thus, to ensure the accuracy, the authors need provide more evidences.

Reviewer 2 Report

Ga2Te3-based composite electrode (Ga2Te3-TiO2-C) is introduced, the synthesis method and the sodium storage properties are systematically studied in this manuscript. The reviewer believes that this manuscript can be accepted after minor revision. Several questions are listed as follows:

1. Is the high electronic conductivity of Te beneficial to the rate performance of Ga2Te3-based composite electrode? Is there any relevant experimental data?

2. It is recommended to supplement the experimental data of the electrochemical performance of Ga2Te3-C and TiO2-C anode prepared at the same conditions, and compare them with that of Ga2Te3-TiO2-C composite anode.

3. In Figure 1(a), there are two strong diffraction peaks near 2q=22o and 35o, which is corresponding to the unreacted raw materials or the unexpected phase generated during the synthesis process? Is there any influence of the unknown phase on the chemical performance of Ga2Te3-TiO2-C composite electrode?

4. In Figure 3(c), the capacity of Ga2Te3-TiO2-C(10%) slightly increased until 200 cycles, followed by a gradual decrease. Please explain the variation trend of the capacity.

5. According to the XRD patterns of Figure 4 (a, b), the changes of the diffraction peaks of the phase during the charging and discharge process are not not significant. It is recommended to amplify the regions of interest or replace the XRD patterns.

Reviewer 3 Report

In the review of the research article titled: Ga2Te3-based Composite Anodes for High-performance Sodium-ion Batteries, the article is written very well and the description is also well mentioned. I would like to see this article publish but after some minor modifications which are suggested as follow;

1-      In EDS spectra there are few unknown peaks, are these impurities and is it affecting the overall performance of electrode. Please justify it.

2-      IN EDS spectra, Te is reported 27% but in TEM elemental mapping, there is a disagreement of the distribution of Te as it seems very less. Please provide some compatible EDS to justify the atomic weightage.

3-      From GCD curves, there is a lot poor reversibility observed between first and second cycle. Can you please perform it again and provide new results with better reversibility of electrode.

4-      In Figure 3c, columbic efficiency scale is needed to be adjusted as it is missing from the start.

5-      The equations given as 5 and 6 needed to be enhance with respect to intercalation and deintercalation of ions whether it’s positive or negative.

6-      Provide some SEM images of an electrode which already performed 300 cycles.

7-      Provide ICP before and after successive charge-discharge cycling.

Round 2

Reviewer 1 Report

The authors have fully addressed all my questions, and the manuscript can be accepted now.